# Genome-Wide Identification and Expression Analysis of HSF Transcription Factors in Alfalfa (*Medicago sativa*) under Abiotic Stress

**DOI:** 10.3390/plants11202763

**Published:** 2022-10-19

**Authors:** Jin Ma, Guozhe Zhang, Yacheng Ye, Linxue Shang, Sidan Hong, Qingqing Ma, Yu Zhao, Cuihua Gu

**Affiliations:** 1College of Landscape and Architecture, Zhejiang Agriculture & Forestry University, Hangzhou 311300, China; 2Zhejiang Provincial Key Laboratory of Germplasm Innovation and Utilization for Garden Plants, Zhejiang Agriculture & Forestry University, Hangzhou 311300, China; 3Key Laboratory of National Forestry and Grassland Administration on Germplasm Innovation and Utilization for Southern Garden Plants, Zhejiang Agriculture & Forestry University, Hangzhou 311300, China

**Keywords:** *Medicago sativa*, HSF gene family, expression profile, abiotic stress

## Abstract

Alfalfa (*Medicago sativa*) is one of the most important legume forage species in the world. It is often affected by several abiotic stressors that result in reduced yields and poor growth. Therefore, it is crucial to study the resistance of *M. sativa* to abiotic stresses. Heat shock transcription factors (HSF) are key players in a number of transcriptional regulatory pathways. These pathways play an essential role in controlling how plants react to different abiotic stressors. Studies on the HSF gene family have been reported in many species but have not yet undergone a thorough analysis in *M**. sativa*. Therefore, in order to identify a more comprehensive set of *HSF* genes, from the genomic data, we identified 16 members of the *MsHSF* gene, which were unevenly distributed over six chromosomes. We also looked at their gene architectures and protein motifs, and phylogenetic analysis allowed us to divide them into 3 groups with a total of 15 subgroups. Along with these aspects, we then examined the physicochemical properties, subcellular localization, synteny analysis, GO annotation and enrichment, and protein interaction networks of amino acids. Finally, the analysis of 16 *MsHSF* genes’ expression levels across all tissues and under four abiotic stresses using publicly available RNA-Seq data revealed that these genes had significant tissue-specific expression. Moreover, the expression of most *MsHSF* genes increased dramatically under abiotic stress, further validating the critical function played by the MsHSF gene family in abiotic stress. These results provided basic information about MsHSF gene family and laid a foundation for further study on the biological role of MsHSF gene in response to stress in *M. sativa*.

## 1. Introduction

Normal plant growth is often affected by a variety of adverse environmental factors such as drought, high salinity, temperature extremes and other abiotic stresses [1]. Abiotic stress inhibits normal plant growth, development, and function by speeding up chlorophyll degradation, disrupting chloroplast membrane activities, and decreasing photosynthetic efficiency, as shown by a number of studies [2,3,4]. In response to external stimuli, plants’ bodies produce signals that trigger the phosphorylation of downstream proteins, which in turn trigger a number of transcription factors [5]. Ultimately, plant-associated resistance genes and associated defense systems are induced, thus altering the ability of the plant to adapt to its environment [6,7,8].

Plants have developed multiple defense mechanisms and strategies to cope with adverse conditions and respond accordingly [9,10,11]. Under abiotic stress, induction of numerous proteins, including transcription factors (TF), can regulate the expression of specific functional genes and enhance plant resistance through signal transduction pathways [12]. Reactive oxygen species (ROS) scavenger enzymes and HSP are important functional proteins induced by HS, and their corresponding genes are targets of several HS-responsive TFs [13]. Previous studies have shown that HSFA6b is essential in *Arabidopsis thaliana* as a downstream regulator of the ABA-mediated heat stress response (HSR) [14,15].

HSF is a class of transcription factors that are widely present in eukaryotes [16]. Since the first *HSF* gene was isolated from *Solanum lycopersicum* in 1990, HSF has been reported in *A. thaliana*, *Oryza sativa*, *Glycine max* and other plants with the continuous improvement of genome sequencing technology [17,18,19,20,21]. The number of HSF gene family members varies widely among plants, with the highest number in *Triticum aestivum* containing 56 [22]. *G. max* [23], *Zea mays* [24], and *A. thaliana* [25] contained 52, 30, and 21, respectively. HSF plays a crucial role in the transmission and receipt of signals, the detection of heat shock components, the regulation of downstream genes, and the induction of plant stress responses [26]. The important role of HSF in plant responses to abiotic stresses can be well established.

Alfalfa (*Medicago sativa*) is one of the most economically valuable crops in the world and the most widely farmed legume fodder grass [27]. However, as it matures and flourishes, *M. sativa* is regularly damaged by a number of abiotic factors, such as salinity, cold and drought, which have a detrimental effect on *M. sativa* quality and output [28,29,30,31]. Therefore, it is particularly important to breed *M. sativa* germplasm resources with high resistance to stress. HSF is a class of transcription factors that are widely found in eukaryotes. They are crucial in the transmission and receipt of signals, the identification of heat shock components and the control of downstream genes, and the induction of plant responses to abiotic stresses [26,32,33,34]. Studies on the HSF gene family have been reported in many species but have not yet undergone a thorough analysis in *M. sativa*. Therefore, in this study, we identified the MsHSF family gene in *M. sativa* by integrating conserved motifs, gene structure, chromosome mapping, promoter cis-elements, and their phylogenetic relationships, and we analyzed the expression of HSF genes in *M. sativa* under four abiotic stresses. These results will deepen our current understanding of the evolutionary relationships and functional differentiation of *M. sativa HSF* genes and provide valuable information for further studies on the role of HSF genes in *M. sativa* for resistance under abiotic stresses.

## 2. Results

### 2.1. Identification of HSF Genes in M. sativa

In this study, we used the *A. thaliana* HSF protein (AtHSF) as a query to search the *M. sativa* genome database for 25 *MsHSF* candidate genes. We then used the HMMER3 search to search the *M. sativa* genome database using the HSF-type DBD model to retrieve 22 putative *MsHSF* candidate genes (PF00447). Finally, we used SMART and NCBI conserved structural domains to delete duplicate genes and proteins without DBD conserved structural domains to obtain 16 MsHSF family members. Based on these genes’ chromosomal positions, they were given the new names *MsHSF01*–*MsHSF16*.

The predicted physicochemical properties of the amino acid sequences indicated that the 16 *HSF* genes encode proteins containing 211 (MsHSF06) to 543 (MsHSF09) amino acids with molecular weights (MWs) ranging from 24,541.86 (*MsHSF06*) to 60,982.41 Da (MsHSF06), with an average molecular weight of 42,922.38 Da. The predicted isoelectric points (pI) ranged from 4.72 (MsHSF05) to 8.16 (MsHSF06), with a mean value of 5.88. Instability index calculations predicted that 14 (87.5%) of the HSF proteins were unstable in vitro, and only MsHSF08 and MsHSF12 predicted proteins with an instability index of less than 40 were classified as stable proteins. In addition, the aliphatic amino acid index (A.I.) ranged from 62.79 (MsHSF08) to 83.61 (MsHSF12), indicating that their thermal stability was less different. The overall mean of the hydrophilic (GRAVY) scores of all HSF proteins was negative, indicating that they are all hydrophilic proteins. Finally, subcellular localization predictions showed that all MsHSF proteins were located in the nucleus (Table 1).

### 2.2. Phylogenetic Analysis of HSF Genes in M. sativa

We created a NJ phylogenetic tree using MEGA6.0 for the amino acid sequences of 16 *M. sativa* HSF, 21 *A. thaliana* HSF, and 25 *O. sativa* HSF in order to study the evolutionary relationships of *MsHSF* genes (Figure 1). Based on the well-established *A. thaliana* family classification, HSF are clearly classified into three groups, namely HSF A (green), HSF B (yellow) and HSF C (blue). Members of MsHSFs were identified in all three groups. The largest was MsHSF A, which made up 52.2% of the overall MsHSF, which was broken down into 9 subgroups (A1–A9). HSF A consists of nine proteins, namely MsHSF01, MsHSF04, MsHSF05, MsHSF07, MsHSF09, MsHSF10, MsHSF11, MsHSF12 and MsHSF16. MsHSF proteins were not aggregated into the three subgroups, A4, A7 and A9. MsHSF B was divided into 5 subgroups (B1–B5), accounting for 39.1%, and consisted of 6 members (MsHSF02, MsHSF03, MsHSF06, MsHSF08, MsHSF13, and MsHSF15). MsHSF C was the smallest group. Also, it contained only one MsHSF14. *M. sativa* HSF proteins clustered more closely with *A.thaliana* HSF proteins, according to the phylogenetic tree, which is interesting. This finding shows that the HSF proteins of dicotyledons and monocotyledons have distinct evolutionary differences.

### 2.3. Multiple Sequence Alignment and Protein Modeling Analysis of the HSF Gene in M. sativa

To check for the presence and position of conserved protein structural domains, we used Jalview to perform a multiple sequence alignment on 16 MsHSF proteins. We found that the DBD structural domain, which contains roughly 100 amino acids, is substantially conserved among all members of the MsHSF family (Figure 2). The DBD structural domain has three helix bundles (1–3) and four reverse parallel folds, as predicted from protein secondary structure (1–4). Additionally, we predicted the protein 3D structure of HSF of *M. sativa* and labeled the starting position of the 1 DBD structural domain in the figure (Figure 3).

### 2.4. Gene Structure and Conserved Motif Analysis of the HSF Gene in M. sativa

HSF proteins were split into three categories, HSF A, HSF B, and HSF C, based on the phylogenetic analyses mentioned above (Figure 4A). Our analysis of the MsHSF gene structure revealed that *HSF* genes from the same group typically have a similar number of introns in their structures. There is just one intron and two exons shared by all *HSFB* and *HSFC* genes in *M. sativa* (Figure 4B). Except for *MsHSF12*, which had eight introns, the *M. sativa HSF A* genes ranged in intron count from one to four. Three genes included two introns; three genes (*MsHSF01, MsHSF07*) contained four introns; two genes (*MsHSF10, MsHSF11*) contained one intron. *MsHSF09* and *MsHSF016* contain four introns.

We also used MEME to identify up to 10 highly conserved motifs in each HSF protein (Figure 4C). The results show that the relative positions of most of the sequence motifs of the same group are similar. All MsHSF contain motifs 1 and 3, which constitute the most highly conserved part of the DBD, and the absence of motif 2 in some genes indicates that motif 2 is not necessary in the highly conserved part of the DBD. Motifs 7 and 9 were found only in HSF A, and motif 8 was present only in HSF B. These results suggest that the biological functions of the MsHSF proteins that are grouped together may be similar, and that different patterns may be related to different functions in separate subgroups.

### 2.5. Chromosome Distribution and Covariance Analysis of HSF Genes in M. sativa

*M. sativa HSF* genes were unevenly distributed on the six chromosomes (Figure 5). This included chromosomes 1, 3, 4, 5, 6, and 7, with the largest number of genes found on chromosome 6 with 5 genes, followed by 3 genes found on chromosome 4. All the remaining chromosomes contained only 2 *MsHSF* genes each.

Gene duplication events are common in all species; they can generate new functional genes and drive species evolution. Therefore, we used MCScanX genomic homozygosity analysis to explore duplications in the *M. sativa* HSF gene family (Figure 6). Three (*MsHSF10*, *MsHSF11*, and *MsHSF12*) tandem duplicated genes and two pairs (*MsHSF01*, *MsHSF02*, *MsHSF15*, and *MsHSF16*) of codominant genes were detected in the MsHSF family.

Additionally, the proportion of *HSF* genes shared by *M. sativa* and other species may reflect the evolutionary relationships of the HSF gene family among *M. sativa*, dicotyledons, and monocotyledons. Therefore, in this study, we constructed a comparative homozygous map of three *HSF* genes from *M. sativa* and three dicots (*A. thaliana*, *G. max*, and *Medicago truncatula*) and one monocot (*Z. mays*) (Figure 7). A total of 44 direct homologous pairs (including one *M. sativa* gene corresponding to more than one *G. max* gene) were studied between *M. sativa* and *G. max HSF* genes. 20, 11, and 7 pairs of direct homologous genes were presented in *M. truncatula*, *A. thaliana*, and *Z. mays*, respectively. During evolution, most of the *HSF* genes of *M. sativa* have more than two direct homologs in *A. thaliana*, which further suggests that *M. sativa* has experienced more whole gene duplication events.

### 2.6. Analysis of the Promoter Cis-Element of the HSF Gene in M. sativa

In order to learn more about how the *MsHSF* gene is regulated, this study examined the cis-acting components of the promoter region. We searched the PlantCARE database for potential cis-acting elements in the 2000 bp upstream sequence of the *MsHSF* translation start codon (Figure 8). As a result, most are stress response elements, and all promoters contain at least one TATA-box, CAAT-box, and short_function element. The next most common element is the ABRE element (contained in all promoters except *MsHSF11*), which is associated with the abscisic acid response. Other elements present in MsHSF promoters include the MBS element (involved in drought, high salt, and low temperature responses), TC-rich repeat sequences (involved in defense and stress responses), AE-box, and TGA erythromycin response elements. These findings suggest that *MsHSF* genes may be involved in multiple transcriptional regulatory mechanisms of plant growth and stress responses.

### 2.7. Expression Profiling of the M. sativa HSF Gene in Different Tissues

To determine the expression patterns of individual *MsHSF* genes in various tissues, a hierarchical clustering heat map was also constructed from public RNA-seq data obtained from NCBI for exploring the transcriptional patterns of *MsHSF* genes in this study (Figure 9). The blue color in the graph indicates low transcript abundance and the red color indicates high transcript abundance. The analysis showed that there were different *HSF* genes showing a trend of high expression in six tissues of *M. sativa*. *MsHSF06* and *MsHSF15* were highly expressed in the roots. The expression of each gene was not significant in the root nodules. In pre-elongating stems, *MsHSF05*, *MsHSF09* and *MsHSF14* showed higher expression. However, the gene with higher expression in elongating stems was *MsHSF03*. In *M. sativa* leaves, *MsHSF07*, *MsHSF08* and *MsHSF16* showed higher expression. Interestingly, *MsHSF10* was not expressed in any other tissue parts of alfalfa but was expressed in the flowers, a phenomenon that suggests that different *MsHSF* genes have more obvious specificity in different tissues.

### 2.8. Expression Analysis of HSF Genes in M. sativa in Response to Abiotic Stresses

To investigate the expression levels of *MsHSF* genes under abiotic stress, we analyzed the expression patterns of *MsHSF* genes under treatment with cold stress, abscisic acid (ABA), drought, and salt using published transcriptome data (Figure 10). The results of the analysis showed that the genes that functioned in *M. sativa* under different stress treatments were slightly different compared to the control. However, the genes that appeared to be the first to function and to be highly expressed in the immediate period of abiotic stress were *MsHSF09* and *MsHSF15*, while *MsHSF07* and *MsHSF08* gradually became more highly expressed as the time of stress increased to counteract the external stimuli. By the later stages of stress, it is *MsHSF04*, *MsHSF05* and *MsHSF13* that play a role. Interestingly, most of the *MsHSF* genes were induced to increase in response to abiotic stress, except for *MsHSF02* and *MsHSF16*, which were suppressed in response to cold stress, which may be related to cellular trauma in *M. sativa* during cold stress.

### 2.9. GO Annotation and Enrichment Analysis of M. sativa HSF Protein

Plants have evolved complex mechanisms to sense and respond to biotic and abiotic stresses, and HSF is an important component of these defense systems. We carried out GO annotation and enrichment analysis on 16 MsHSF proteins in order to learn more about the biological functions of this protein (Figure 11). MsHSF was enriched for 55 biological processes, 1 cellular component, and 2 molecular functions in comparison to the entire GO database. According to the GO enrichment data, MsHSF transcription factors are primarily involved in biological processes including responding to abiotic stimuli, responding to temperature stimuli, responding to heat, and responding to xenobiotic stimuli. The findings again suggest that HSF genes play an extremely important role in resisting abiotic stresses.

### 2.10. Interaction Network Analysis of HSF Proteins in M. sativa

We predicted probable interactions between MsHSF proteins using the STRING database in order to better comprehend MsHSF protein interactions (Figure 12). The findings demonstrate that while certain proteins, like MsHSF10 and MsHSF16, exhibit direct connections, others, including MsHSF10, MsHSF16, and MsHSF08, exhibit more complex multigene interactions. where it is projected that the major nodes MsHSF01, MsHSF08, and MsHSF13 radiate a significant number of connections to additional nodes.

## 3. Discussion

### 3.1. The Characteristics of HSF Gene Family in M. sativa

HSF is a particular sort of transcription factor that is crucial for plants’ ability to resist diverse stressors [35]. The highly conserved plant HSF DBD is located at the N-terminal end where it is able to precisely locate and recognize the heat stress element (HSE) in the promoter of the target gene [36,37,38]. All MsHSF proteins comprised DBD with three helices and four folds, according to multiple sequence alignment and secondary structure prediction (Figure 2). It’s interesting to note that some MsHSF proteins also have additional conserved structural domains; further experimental confirmation is required to determine whether this is a sign of the gene family’s functional diversification. Tertiary structural analysis showed that the portion of transcription factors interacting with nucleic acids is conserved in the subfamily. HSF B and HSF C members may not have transcriptional activation because they lack AHA motifs. which is consistent with the results of previous studies [39,40,41,42].

Similar gene architectures and conserved protein motifs among members of the same phylogenetic group typically indicate a tight phylogenetic relationship [43]. Short sequences involved in significant biological processes are typically referred to as motif [44]. The presence of Motifs 1–2 in every MsHSF raises the possibility that they may have significant biological roles, however this has not yet been established. Because motif 9 is specific to the HSF A subgroup, it is possible that the *HSF* genes in this subgroup perform a particular role. Additionally, we discovered that the majority of MsHSF share exon-intron architectures and motif distributions within the same evolutionary tree grouping, indicating that genes within a subfamily frequently have comparable biological activities.

Only 16 HSF genes have been found in *M. sativa*, which is less than other plant species and may reflect the lack of expansion of the MsHSF family. MsHSF genes might be further grouped into three categories using homology matching and multispecies matching: A, B, and C. They are clustered in the same way as the members of the *A. thaliana* HSF gene family, with group A having the most genes and group C having the fewest [26]. Three subgroups, A4, A7, and A9, are missing from *M. sativa*, demonstrating that despite HSF family proteins sharing a common ancestor, they have evolved separately in different species. The majority of the alfalfa HSF proteins grouped with the *A. thaliana* HSF proteins but not with the *O. sativa* HSF proteins, suggesting that MsHSF and AtHSF have a tight evolutionary relationship. The HSF proteins of dicotyledons and monocotyledons have evolved in quite different ways.

Gene duplication events have a big impact on how gene families are formed. By supplying the necessary building blocks for the creation of new genes, gene duplication aids in the development of new, functional genes [45]. The majority of gene duplication occurs as tandem and fragmental duplication [46]. The 16 *MsHSF* genes in the *M. sativa* genome contained four homologous gene pairs, all of which underwent WGD (whole genome duplication) or fragmental duplication events and intense purifying selection pressure. These findings imply that WGD or fragmental replication is essential for *MsHSF* gene amplification.

Cis-acting elements are nucleotide sequences found upstream or downstream of genes that regulate their transcriptional levels [47]. When plants react to numerous developmental processes and stressors, they work by binding to certain transcription factors [48]. According to studies, cis-acting elements are present in plant-inducible promoters in response to adverse stress. There are numerous hormone-responsive core promoter elements and binding sites spread across the 16 *M. sativa* HSF promoter regions. According to this, MsHSF might be involved in the communication between several hormone signaling pathways. Aside from the heat stress element, the majority of MsHSF also contained the drought response element MBS, the anaerobic induction response element ARE, and the low temperature response element LTR. This shows that this gene family may control the effects of a variety of abiotic stimuli. These findings imply that MsHSF may interact with hormone signaling pathways that control growth and development as well as stress responses in *M. sativa*.

According to GO enrichment analysis, 15 of the 16 *MsHSF* genes are involved in two biological processes of GO resistance to abiotic stress and synthesis of abiotic stress factors (Figure 11). Several studies have shown that HSF regulates the expression of stress-related proteins, such as heat shock protein (HSP), which plays an important role in the plant stress response, especially heat stress [49,50,51]. Therefore, we speculate that the *MsHSF* gene may play a key role in plant resistance to abiotic stresses.

### 3.2. The Potential Roles of Differentially Expressed MsHSF Genes

In order to protect plants from heat stress, HSPs can raise the denaturation temperature of their proteins. They can also fix damaged proteins, enabling plants to withstand high temperatures [52]. HSFs play a major role in transcriptionally controlling the expression of HSPs. Additionally, the role of the HSFs signaling pathway encompasses many stresses, including cold, osmosis, drought, and salt, in addition to the response to heat stress [53]. It’s interesting to note that osmotic pressures such as drought, salinity disruption, and other stresses result in the buildup of ROS, ABA, and H_2_O_2_ as well as alterations to cell walls. Ca^2+^ and ROS are the key factors causing abiotic stress response processes. Therefore, we examined the *M. sativa* transcriptome under ABA, salt, drought, and low temperature stressors. It was discovered that the expression of a considerable number of HSF genes was elevated under these stressful circumstances. This suggests that these *HSF* genes may be involved in some processes in the response of plants to external stresses. In conclusion, *MsHSF* genes are an important class of regulatory genes that control the plant’s growth, development, and response to stress.

HSF plays an important role in the plant’s response to abiotic stresses because it can achieve resistance to abiotic stresses by regulating the expression of different genes [54,55]. Among the *HSF* genes in plants, HSFA is a major transcriptional activator because it is essential to awaken HSR [56,57,58]. Although there were some subtle differences in the responses of *MsHSF* genes to different stresses in this study, the first to play a role in resistance was the *MsHSF09* gene in HSFA, which reinforces the important role of HSFA in resistance to abiotic stresses. Unlike HSFA, a considerable number of HSFB and HSFC have not been reported as transcriptional activators, but interestingly, in *M. sativa*, many genes in HSFB, such as *MsHSF07*, *MsHSF08*, and *MsHSF15*, also play important roles in resistance to abiotic stresses, and based on previous studies and analyses, it is clear that genes in HSFB in alfalfa function as transcriptional co-activators of HSFA. Taken together, several MsHSF genes are differentially expressed under abiotic stresses (including heat, salt, or ABA stress), and these results suggest that they may be involved in plant responses to abiotic stresses.

## 4. Materials and Methods

### 4.1. Identification and Sequence Analysis of HSF Genes in M. sativa

HSF protein sequences from the *A. thaliana* TAIR database (https://www.arabidopsis.org/) (accessed on 7 June 2022) were used as a reference sequence, and members of the putative *MsHSF* gene were sought using *M. sativa* genomic data from the National Genomics Data Center (https://ngdc.cncb.ac.cn/) (accessed on 7 June 2022) and a BLASTP search. We then used the native HMMER 3.0 software (Robert, D.F.; Ashburn, VA, USA) [59], using Hidden Markov (HMM) mapping of the HSF protein (PF0047), downloaded from the pfam database (http://pfam.xfam.org) (accessed on 9 July 2022). The potential gene members of *M. sativa* HSF identified were pooled using these two search techniques. WebCD-search (https://www.ncbi.nlm.nih.gov/cdd) and SMART (http://smart.embl.de/) were used to identify conserved HSF structural domains in all potential *MsHSF* genes (accessed on 9 July 2022). We finally identified 16 *MsHSF* genes and renamed them according to their position on the *M. sativa* chromosome.

We predicted and examined the physicochemical characteristics of all MsHSF potential proteins, including amino acid numbers, molecular weights, and theoretical sites, through the website ExPasy (https://www.expasy.org/) (accessed on 11 August 2022). Cell-PLoc2.0 was used to create subcellular localizations (http://www.csbio.sjtu.edu.cn/bioinf/plant-multi/#) (accessed on 11 August 2022).

### 4.2. Construction of Phylogenetic Tree and Sequence Comparison

Whole genome information for *A.thaliana* and *O.sativa* was downloaded from the NCBI database (https://www.ncbi.nlm.nih.gov/) (accessed 2 August 2022). Using MUSCLE technology, the 16 MsHSF protein sequences, 21 AtHSF proteins, and 25 OsHSF sequences found were compared to multiple sequences. The comparison parameters were in multiple comparison mode (other parameters were in default mode), and the obtained comparison results were used to construct a neighbor-joining (NJ) phylogenetic tree that we created in MEGA 7.0 using 1000 bootstrap replications, and the phylogenetic tree was created as an illustration using iTOL (https://itol.embl.de/) (accessed on 2 August 2022).

Intraspecific classification of *M. sativa* HSF sequences was performed based on interspecific phylogenetic trees, and amino acid sequences of conserved structural domains were compared and modified using Jalview software 2.11.2.4 (Andrew, M.W.; Cambridge, MA, USA) (http://www.jalview.org/) (accessed on 2 August 2022), and conserved thematic WebLogo were produced.

We also submitted the Jalview output to SOPMA (https://npsa-prabi.ibcp.fr/cgi-bin/npsa_automat.pl?page=npsa_sopma.html) (accessed on 3 August 2022) for protein secondary structure prediction by using default parameters. For tertiary structure prediction, we completed the analysis via the online website SWISS-MODEL (https://swissmodel.expasy.org/interactive) (accessed on 3 August 2022).

### 4.3. Gene Structure and Motif Identification

The conserved amino acid sequences of HSF proteins were analyzed using the online MEME tool (https://meme-suite.org/meme/) (accessed on 5 August 2022) with the parameters of minimum width of ≥ for 6, maximum width of 50, and number of parentheses of 10; all other parameters were set to default values. Then we obtained the intron-exon distribution of the *MsHSF* gene from the *M. sativa* genome using the GFF annotation method, and finally the results were displayed using TBtools software v1.098661 (Chen, C.J.; Guangzhou, China) [60].

### 4.4. Chromosome Location and Covariance Analysis

Chromosome lengths and gene positions were obtained from *M. sativa* genome annotation files, and MG2C v.2 (http://mg2c.iask.in/mg2c_v2.0/) (accessed on 8 August 2022) was used to visualize gene positions on chromosomes. *M. sativa* protein sequences were aligned to each other or to those of *A. thaliana*, *G. max*, *Z. mays*, or *M. truncatula* using TBtools software. MCScanX was used along with default parameters to identify homozygous relationships between gene replication events and HSF proteins, and results were visualized using Circos and Dual Synteny Plot in TBtools.

### 4.5. Identification of Cis-Acting Elements

A 2000 bp promoter region upstream of the MsHSF transcriptional start site was extracted from the *M. sativa* genome (http://www.genoscope.cns.fr/brassicanapus/) (accessed on 25 August 2022) and submitted to PlantCARE (http://bioinformatics.psb.ugent) (accessed on 25 August 2022) to identify the three types of regulatory cis-acting elements and finally visualize the results using TBtools.

### 4.6. Analysis of Tissue-Specific Expression and Abiotic Stress Transcriptome Data

Relevant transcriptome data were downloaded from the public NCBI database to investigate the expression patterns of *MsHSF* genes in different tissues and under different biotic stresses. *M. sativa* RNA-Seq data for different tissues (pre-elongated stems, elongated stems, flowers, leaves, roots, and rhizomes) can be downloaded from accession number SRP055547. The NCBI Short Read Long Archive database contains abiotic stress-related transcriptome data (cold, at SRR7091780-SRR7091794; drought, salt, and ABA, at SRR7160313-SRR7160357). The raw data was filtered and the SRA files were converted to FASTQ files by the SRA to Fastq program of TBtools. Gene expression levels were calculated by fragment number per kilobase per million mapped reads (FPKM values). Finally, the Hetmap program of TBtools was imported to generate the associated heat map.

### 4.7. Protein Interaction Network Prediction and GO Enrichment Analysis

HSF protein sequences were uploaded to the STRING database (https://string-db.org/) (accessed on 19 August 2022) for node comparison, and relationships between important proteins were predicted based on *A. thaliana* protein interactions. Finally, Cytoscape (Shannon, P.; California, USA) [61] was used to visualize the generated networks.

The online software EggNOG-Mapper (http://eggnog-mapper.embl.de/) (accessed on 18 August 2022) was used to annotate the GO function of the *MsHSF* gene. The results were collated using the eggNOG-mapper Helper function of TBtools and the text files used for downstream analysis were exported separately to GO Enrichment for enrichment analysis. Finally, use the online charting tool HIPLOT (https://hiplot.com.cn/) (accessed on 19 August 2022) to view and examine the data.

## 5. Conclusions

From the genomic information of *M. sativa*, we discovered a total of 16 *MsHSF* genes in this study. We analyzed the physicochemical properties of the 16 MsHSF proteins and found that all the *MsHSF* genes were localized in the nucleus. Amino acid sequence comparison showed that all *MsHSF* genes contain conserved DBD structures, after which we classified MsHSF proteins into 2 groups and 15 subgroups based on evolutionary relationships and showed that they are mostly similar within the same group but differ significantly between subgroups. Cis-acting element and GO enrichment analyses suggest that *MsHSF* genes may be involved in multiple transcriptional regulatory mechanisms for plant growth and stress response. In addition, expression profiling indicated that *MsHSF* genes could show significant specificity during tissue development and that *MsHSF* genes play a very important role in response to abiotic stresses. Overall, bioinformatic analysis and expression profiling studies of HSF can help to understand the important role of HSF in abiotic stress responses in alfalfa and provide a basis for exploring ways to understand and regulate these stress responses.

## Figures and Tables

**Figure 1 plants-11-02763-f001:**
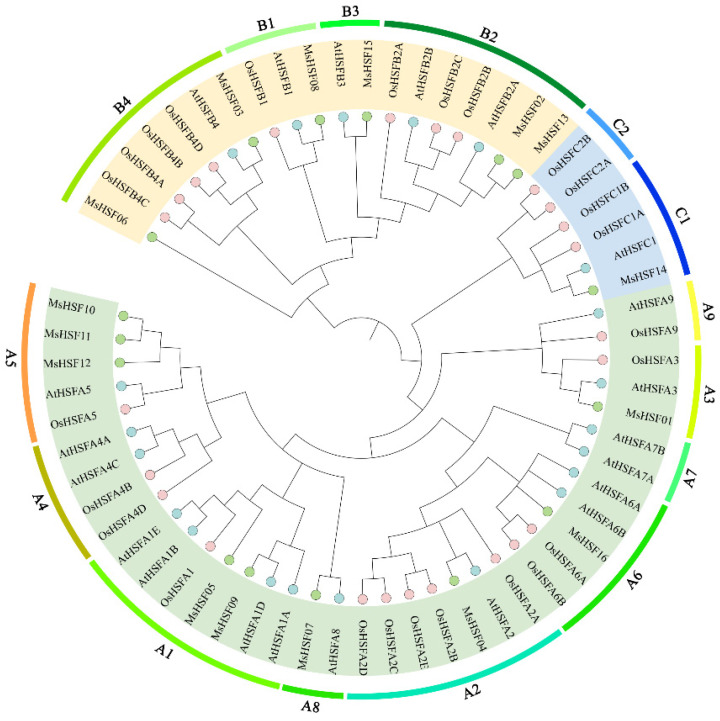
Model for phylogenetic analysis of HSF in *M. sativa*. Each subgroup is distinguished by a different color.

**Figure 2 plants-11-02763-f002:**
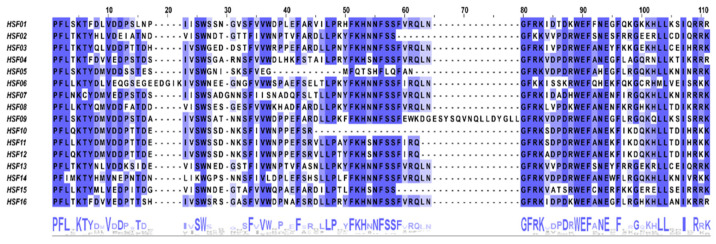
Conserved domain alignment of MsHSF members.

**Figure 3 plants-11-02763-f003:**
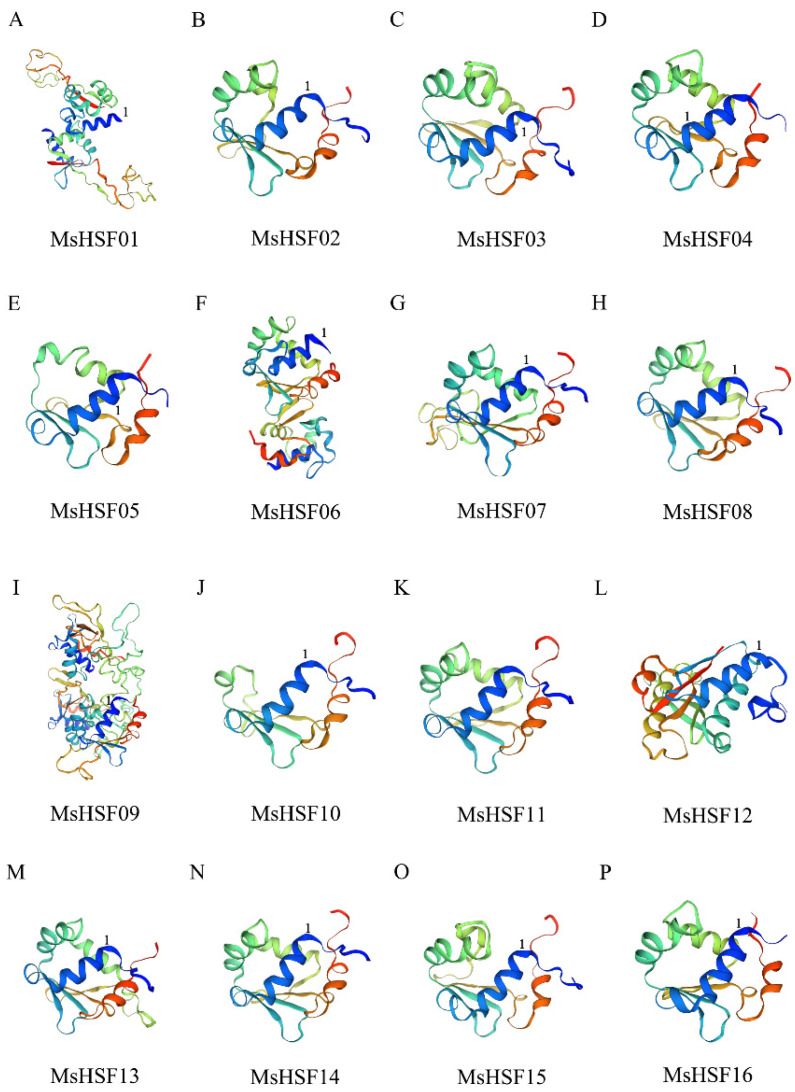
Three-dimensional model of the protein of MsHSF family members, with the starting position of the α1 DBD structural domain indicated by 1. (**A**–**P**) in the figure represent the 16 proteins of the *M. sativa* HSF family, respectively.

**Figure 4 plants-11-02763-f004:**
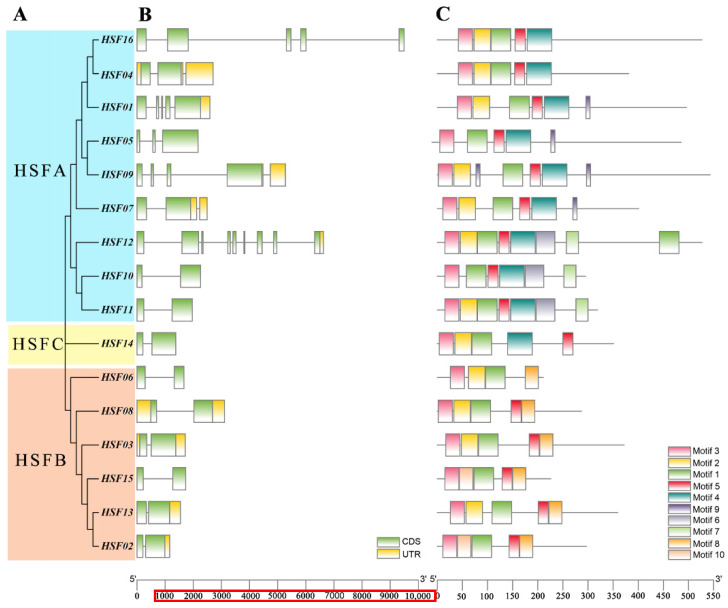
Phylogenetic relationship tree (**A**), gene structure (**B**) and conserved patterns (**C**) of HSF in *M. sativa*.

**Figure 5 plants-11-02763-f005:**
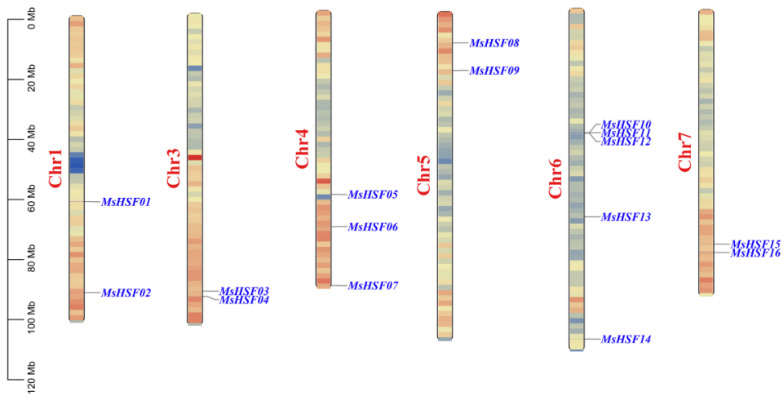
Distribution of *MsHSF* genes on the chromosomal scaffolds of *M. sativa*.

**Figure 6 plants-11-02763-f006:**
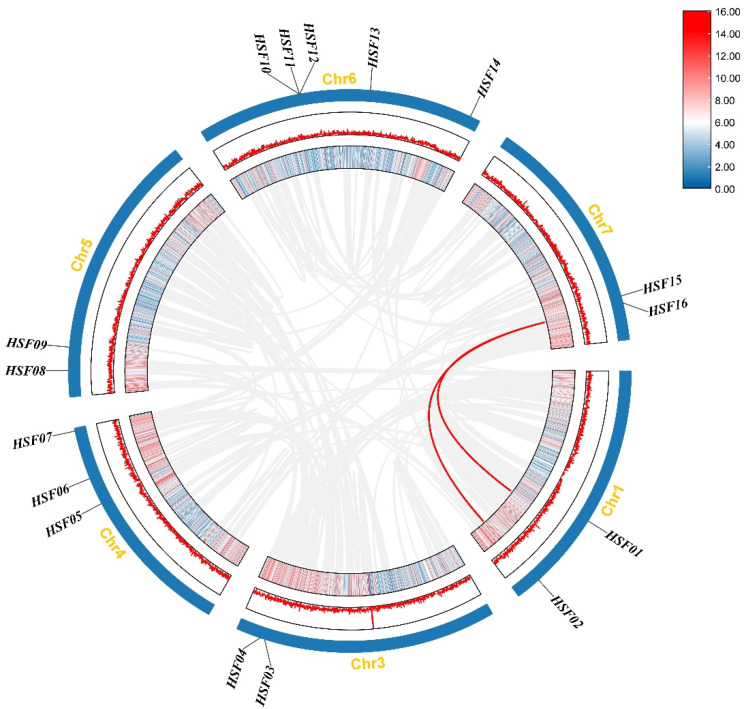
Chromosome distribution and interchromosomal relationships of *MsHSF* genes. Gray lines indicate synthetic blocks within the *M. sativa* genome, and red lines indicate duplicated *MsHSF* gene pairs.

**Figure 7 plants-11-02763-f007:**
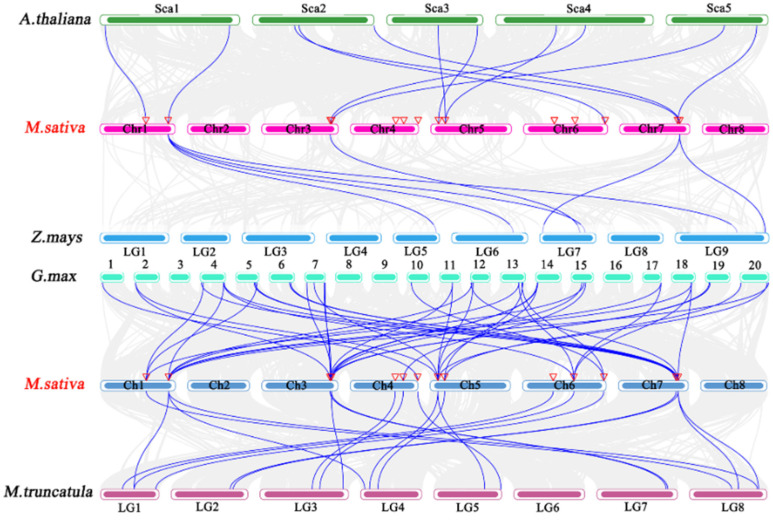
Synthetic analysis of the *M. sativa* genome with the genomes of one monocotyledon and three dicotyledon plants. Gray lines represent alignment blocks between paired genomes, and blue lines indicate synthetic *HSF* gene pairs.

**Figure 8 plants-11-02763-f008:**
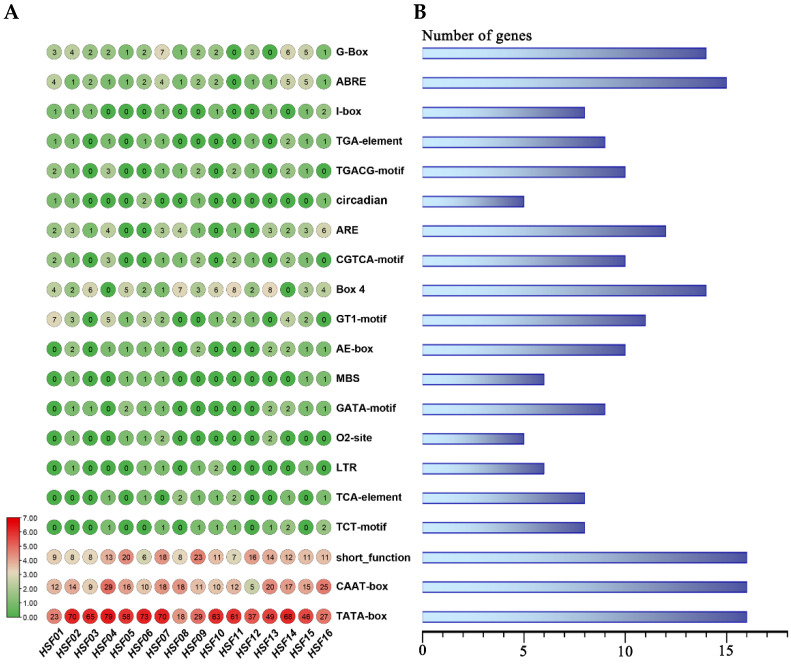
A schematic representation of the cis-acting elements identified in the 2000 bp promoter region upstream of the *MsHSF* gene. The different colors represent the number of cis-acting elements contained (**A**). The number of *MsHSF* genes corresponding to the cis-acting element (**B**).

**Figure 9 plants-11-02763-f009:**
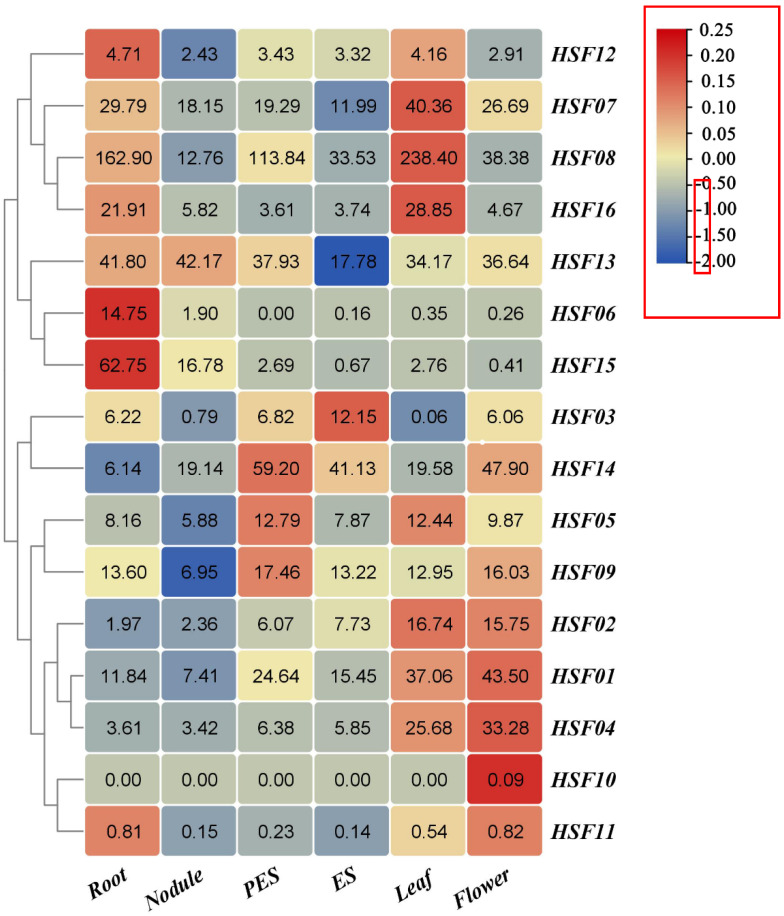
A heat map representation of *MsHSF* expression between different tissues. The values in the rectangle represent the magnitude of the gene expression.

**Figure 10 plants-11-02763-f010:**
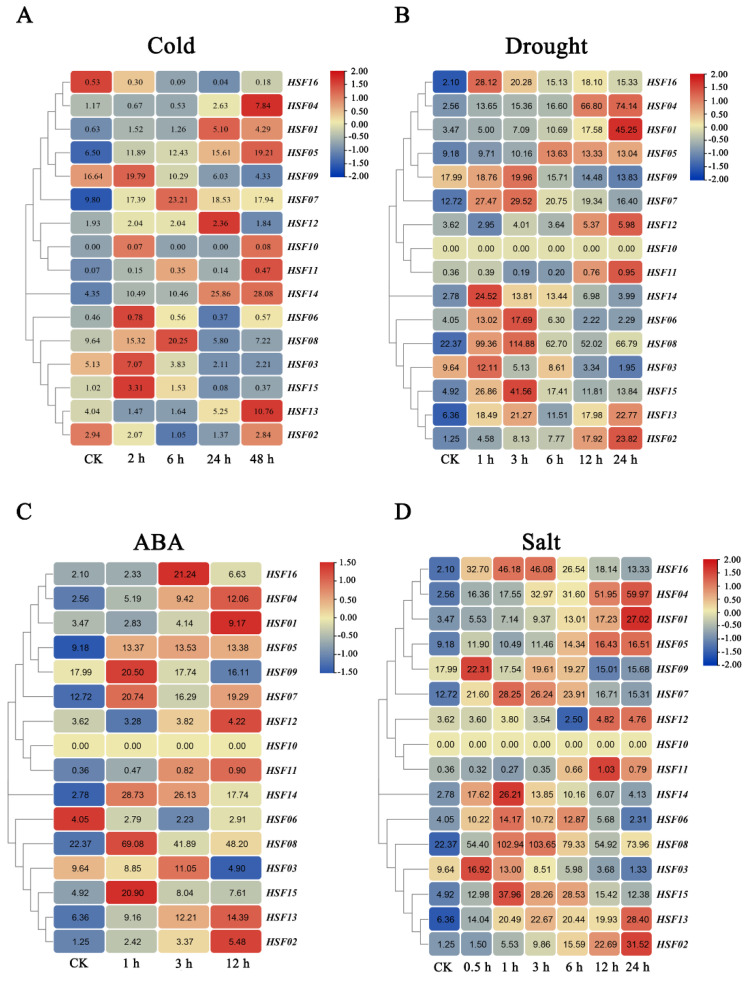
Expression of 16 *MsHSF* genes in cold (**A**), drought (**B**), ABA (**C**), and salt (**D**) treatments. The values in the rectangle represent the magnitude of the gene expression.

**Figure 11 plants-11-02763-f011:**
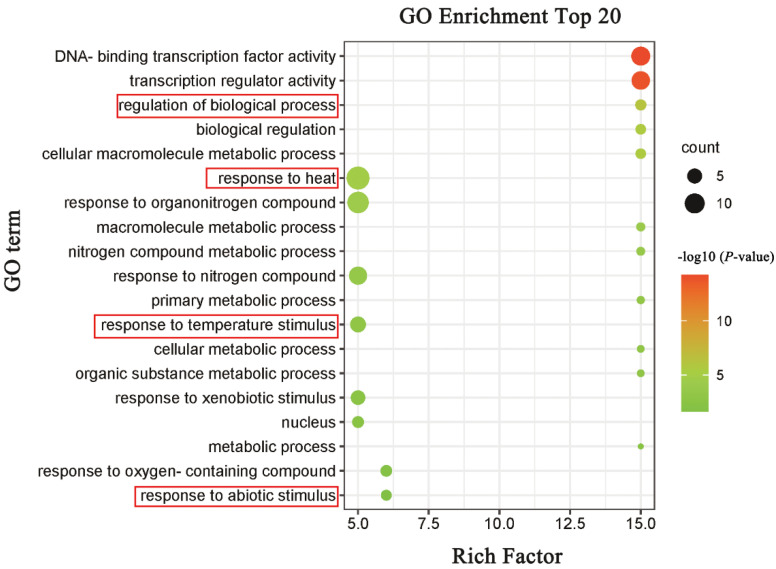
GO enrichment analysis of the MsHSF proteins relative to the GO database. The horizontal axis indicates the enrichment factor, and the size of the circle indicates the number of genes annotated with a given GO term.

**Figure 12 plants-11-02763-f012:**
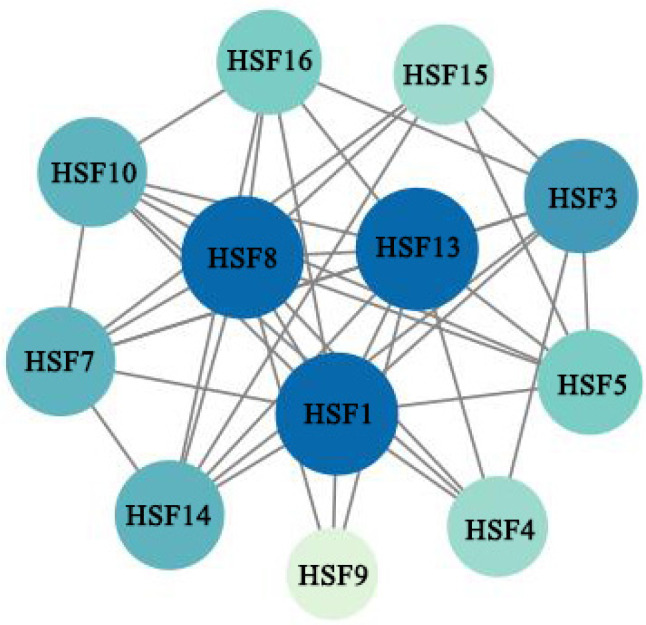
Interaction network of HSF proteins in *M. sativa*. Nodes represent proteins; central nodes are indicated in blue, and black lines indicate interactions between nodes.

**Table 1 plants-11-02763-t001:** Prediction of physicochemical properties of MsHSF.

Protein Name	Gene ID	Chromosome Location	Size (aa)	MW (kDa)	pI	Stability	A.I	GRAVY	Predicted Location
HSF01	MsG0180003410.01.T01	S1:61939996–61942587	496	55,750.22	4.91	U	69.91	−0.591	Nuclear
HSF02	MsG0180005479.01.T01	S1:92250440–92250760	297	32,972.9	6.26	U	71.28	−0.613	Nuclear
HSF03	MsG0380017150.01.T01	S3:92635955–92637670	372	42,094.06	8.16	U	62.82	−0.715	Nuclear
HSF04	MsG0380017281.01.T01	S3:94430904–94433609	381	42,838.88	4.93	U	78.5	−0.547	Nuclear
HSF05	MsG0480021648.01.T01	S4:61250000–61252171	480	53,424.6	4.85	U	69.62	−0.55	Nuclear
HSF06	MsG0480022400.01.T01	S4:72054039–72055708	211	24,541.86	6.02	U	69.72	−0.708	Nuclear
HSF07	MsG0480023951.01.T01	S4:91618543–91621035	401	45,748.67	4.99	U	80.7	−0.498	Nuclear
HSF08	MsG0580024826.01.T01	S5:10417724–10420833	287	32,168.91	6.9	S	62.79	−0.784	Nuclear
HSF09	MsG0580025495.01.T01	S5:19600912–19606188	543	60,982.41	5.1	U	71.47	−0.624	Nuclear
HSF10	MsG0680032444.01.T01	S6:41477418–41479676	295	33,653.64	5.66	U	68.14	−0.802	Nuclear
HSF11	MsG0680032449.01.T01	S6:41520725–41522690	319	36,439.77	5.27	U	70.34	−0.717	Nuclear
HSF12	MsG0680032451.01.T01	S6:41583472–41590098	527	59,504.68	6.96	S	83.61	−0.474	Nuclear
HSF13	MsG0680033612.01.T01	S6:69476487–69478031	359	39,836.88	5.54	U	76.3	−0.439	Nuclear
HSF14	MsG0680035659.01.T01	S6:110235176−110236554	351	40,233.48	5.58	U	63.85	−0.664	Nuclear
HSF15	MsG0780040480.01.T01	S7:78201225–78202956	226	26,119.55	7.56	U	71.55	−0.755	Nuclear
HSF16	MsG0780040676.01.T01	S7:80984782–80994274	527	60,447.57	5.43	U	73.76	−0.61	Nuclear

## Data Availability

All data in this study can be found in the manuscript.

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
