# Peer review of "Genome-Wide Identification and Expression Analysis of HSF Transcription Factors in Alfalfa (Medicago sativa) under Abiotic Stress"

_plants, 2022, doi:10.3390/plants11202763_

Round 1

Reviewer 1 Report

The aim of the manuscript is to characterise the heat shock transcription factor family in Medicago sativa in details; including sequence comparisons, gene structure identifications, chromosomal locations, tissue and abiotic stress specific expression profiling etc.  This is a purely in silico analysis based on various publicly available data bases and bioinformatics tools, its merit is in organising the diffuse and fragmented results of other research groups into one solid piece of information on this topic. Thus, the manuscript is informative, building on the latest genomics and bioinformatics tools.   

The lists of references however is quite unsatisfactory, as most of the referred papers (there are only 47 papers listed) are not from the literatures of Medicago sativa and even less from the field of any aspects of abiotic stress tolerance of Medicago sativa itself, although this is a vast field. Making just a quick filtering for key words Medicago sativa and heat shock proteins resulted in thousands of scientific articles from 2018 to today.

In addition, there are several minor mistakes that should be dealt with. Lists of corrections to be made in the text:

Line 21: please correct to – “from”

Lines 88-90: It is stated that MsHSF08 and MsHSF12 are considered stable proteins, but then in Table 1 they were marked as U; please clarify.  

Line 103: For better understanding the following section, this part should be started with a sentence similar like this: Members of MsHSFs were identified in all three groups; the largest was MsHSF A ….

Lines 104-106: the sentence has no verb in it.

Lines 109-110: As all the three groups (A,B,C) were mixtures from different species, please correct these two sentences to the following: HSF C was the smallest….. Also it contained only one MsHSF….

Line 116: please correct the title to – modelling

Figure 4a: Please indicate the A,B,C groups on the phylogenetic tree (Fig4a)

Figure 6: Gene duplications are indicated in the figure, but two red lines points (from HSF08 and from HSF03, 04) to chrom8, where there was no MsHSF located. Please explain in the text.  

Lines 175-178: Please reconsider this sentence. From the upper part of Fig7 rather the opposite can be seen, that At and Zm show more genome-wide duplications compared to Ms.

Figure 9 and 10: Please clarify in the figure captions, what the numbers in the rectangles stand for.

Line 285: leave out “The” in the middle of the sentence

Line 322: correct to H2O2

Line 323: correct to Ca2+

Line 327: reconsider the use of “body” here

Line 342: Please check the short sentence there “Ensure the correctness of the results.” In its present form, it seems to be unnecessary.

Line 414: put into italics MsHSF genes

Reviewer 2 Report

1.“However, no studies on the HSF transcription factor 66 family in M. sativa have been found so far.” this is not true. There are several studies about HSF in M. sativa. The authors should be more rigorous.  

2.Line 124 “The DBD domain may be able to find and distinguish the heat stress factor with great accuracy, and other plants have displayed comparable behaviour.”what does this sentence mean?

3.Figure 4: This image should be replaced by a higher resolution image.

4.“herefore, in this study, we constructed a comparative homo- 170
zygous map of three HSF genes from M. sativa and three dicots (A. thaliana, G. max, and 171
Medicago truncatula) and one monocot (Z. mays”

Why are these species selected for comparison?

5.2000 bp promoter region upstream of the MsHSF transcriptional start site was ex- 386
tracted from the M. sativa genome

“transcriptional start site” or “start codons”? If it was transcriptional start site, how are these determined?

6. One purpose of this study is to supply candidate HSF genes associated with abotic stress tolerance. Therefore, the discussion should be focused on the possibilty of these 16 genes based on the combined analysis of cis elements, expression patterns and homologous genes in other species. The discussion should be improved.

7.There are plenty of mistakes in writing, and the writing should be improved.

Reviewer 3 Report

Ln 48: The abbreviation of HSF needs to be defined at the first mention in the manuscript.
Ln 53- 54: Incomplete sentence: ' G. max....21, respectively. Correct it.
Ln 56-57: Are you implying that the role of HSF in plant responses to abiotic stress is well established? If so, the phrase 'can be well established' need to be corrected.
Also, this line at the end of the paragraph sounds non-sequitur. I think it should come before the sentence that states the role of HSFs in lines 54-56.
Ln 61: Consider replacing 'output' with 'yield'.
Ln 63: Redundant sentence. Previously stated in line 48
Ln 64-66 Redundant sentence/information. Previously stated in lines 54-56
Ln 68: Define MsHSF at the first mention in the manuscript similar to what you did for AtHSF in line 76.
  The authors should explain why the NJ tree was selected for constructing the Phylogenetic tree. Did the authors compare the other tree construction algorithms such as UPGMA by correlation analysis (Please refer to https://pubmed.ncbi.nlm.nih.gov/32559639/)?

Author Response

请参阅附件。

Round 2

Reviewer 1 Report

The authors have considered the suggestions and chnaged the manuscript accordingly.

Reviewer 2 Report

accepted